# An Assessment of the Climate Change Impacts on the Distribution of the Glacial Relict Woodpecker Three-Toed Woodpecker *Picoides tridactylus*

**DOI:** 10.3390/ani14131879

**Published:** 2024-06-26

**Authors:** Teodora Popović, Nina B. Ćurčić, Snežana Đurđić, Gorica Stanojević, Marko Raković

**Affiliations:** 1Physical Geography Department, Geographical Institute “Jovan Cvijić”, Serbian Academy of Sciences and Arts, Đure Jakšića 9, 11000 Belgrade, Serbia; n.curcic@gi.sanu.ac.rs (N.B.Ć.); g.stanojevic@gi.sanu.ac.rs (G.S.); 2Department of Physical Geography, Faculty of Geography, University of Belgrade, Studentski Trg 3/III, 11000 Belgrade, Serbia; snezana.djurdjic@gef.bg.ac.rs; 3Department of Biology and Inland Waters Protection, Institute for Multidisciplinary Research, University of Belgrade, Bulevar Despota Stefana 142, 11060 Belgrade, Serbia; markorakovic@imsi.bg.ac.rs

**Keywords:** habitat specialist, climate scenarios, species distribution modelling, Balkan Peninsula

## Abstract

**Simple Summary:**

Global warming represents a threat to all boreal plant and animal species on the Balkan Peninsula, such as the Three-Toed Woodpecker *Picoides tridactylus*. As a cold-adapted bird species whose presence on the Balkan Peninsula is limited to mountain forests, certain effects of climate change on its distribution can be expected. Therefore, our aim was to identify the current species distribution and then project the current model into paleoclimate and future climate conditions. Our results show that projected warming over the Balkan Peninsula will have negative impacts on the future distribution of species as there is a contraction of the area with suitable habitats. A cause of concern is the drastic reduction in highly and moderately suitable habitats regardless of the emissions scenario (average loss of 90%). Special attention should be paid to the western part of the Balkans, where the highest percentage of areas with suitable habitats was recorded. Preserving these areas by addressing existing conservation problems could be a way to mitigate the effects of climate change on this species and enable it to adapt more easily.

**Abstract:**

The Three-Toed Woodpecker *Picoides tridactylus* is a rare and endangered woodpecker on the Balkan Peninsula. Despite being widely distributed in Northern Europe, its distribution on the Balkan Peninsula is limited to high-altitude forest habitats, where it represents a glacial relict. Assessing the climate change impacts on its distribution can be crucial for improving the conservation and future survival of this specialist species on the Balkan Peninsula. We used species distribution modelling (SDM) to identify its potential distribution in the past (last interglacial and last glacial maximum), present, and future (2050 and 2070). Our results indicate that this species had the greatest distribution during the last glacial maximum, after which its distribution contracted to areas where suitable environment persisted (high altitudes). The largest territory of the Balkan Peninsula has an unsuitable environment for the species to inhabit, while highly suitable habitats have the smallest share in the total area of suitable habitats. All future models show a decrease in the area of suitable habitats compared with the current period, indicating that global warming has a negative effect on the distribution of the species. We recommend that conservation activities must be of greater extent to ensure the species’ survival in the Balkans.

## 1. Introduction

Throughout the Pleistocene, the Balkan Peninsula played an important role in the survival of many European plant and animal species. During the glacial period, Northern Europe was covered by a large ice sheet, whereas Southern Europe had limited glaciation [1]. The optimal conditions for the survival of trees on the Balkan Peninsula, with the fauna associated with them, were located at mid-altitudes because the lower terrains had a drier climate with steppe vegetation, whereas higher altitudes were characterised by colder conditions [2]. The Balkan Peninsula, with its complex physical-geographic features represented by topographic heterogeneity and numerous microhabitats of a suitable climate, provided favourable environmental conditions for the survival of species that migrated from Northern and Central Europe [2]. At the same time, this variability in relief allowed species to migrate to higher altitudes [3], which became particularly important when the ice sheet began to retreat in approximately 17,000 BCE [4] and the transition from the last glacial maximum to the current interglacial occurred.

As the climate changed, the importance of mountains in the Balkan Peninsula increased for cold-adapted species, such as the Three-Toed Woodpecker *Picoides tridactylus* (Linnaeus, 1758) (Piciformes: Picidae), owing to the possibility of altitudinal movement. The three-Toed Woodpecker is widely distributed in the taiga belt of northern Eurasia, but subspecies like *Picoides tridactylus alpinus* also occur isolated in the mountain belts of Central, South, and Southeastern Europe [5]. This species represents a glacial relict on the Balkan Peninsula [6], where it is more of a boreal-montane bird [7] due to its close relation to high altitudes where the favourable climate is present, in contrast to the boreal characteristics it has in the north. In addition, an important difference between its northern and southern distribution is that, unlike in the north where it is widely distributed [8], the species distribution in the Balkan Peninsula is fragmented [9]. It represents a rare bird species, and some of the parts of the Balkans for which there are records of its occurrence so far are the mountains in Montenegro, Serbia [7,9], Bosnia and Herzegovina [10], Croatia [11], Slovenia [12], and Bulgaria [13]. In the localities mentioned above, the species was observed exclusively in montane forest habitats, typically coniferous, except in Tara Mountain where it occurs in mixed forests.

Mountain ecosystems are particularly sensitive to climate and other environmental changes because of their physical-geographic features (steep relief and climate gradient) and adaptation of species to cold conditions [14,15]. Since the second half of the 20th century, a positive average annual temperature and a negative average annual precipitation trend has been recorded on the Balkan Peninsula [16]. This is in agreement with the significant warming that has been observed in all Europe since the 1960s [17]. According to Peneva et al. [18], the Balkan Peninsula is expected to experience warmer and drier summers, along with increased winter precipitation during the current decade. The Mediterranean, where the Balkan Peninsula is located, is a hotspot of climate change and its impacts [19], with trend rates exceeding global values [20]. According to the IPCC [21], under the worst-case scenario of GHG emissions, the temperature rise over the Mediterranean will range between 3.5 and 8.75 °C by 2100, whereby the warming will be enhanced over the Balkans. These changes will be particularly pronounced in summer, which will have up to 50% higher values compared with the level of global annual warming [21]. A positive temperature trend in both the warmest and coldest months is already being recorded over the peninsula [18]. It is expected that the decrease in average annual precipitation over the Balkans will be up to 40% in the period 2071–2100 compared to the period 1917–2000 [16]. 

Because of their long-term monitoring, birds are a good indicator of climate and other environmental changes [22]. They can respond to climate change by in situ adaptation, migration to other areas with suitable conditions, or extinction if they cannot succeed in any of these [23]. Changes in phenology and changes in distribution are the two major responses already being recorded [24]. Species with a narrow distribution and species specialised to specific environmental conditions are particularly vulnerable [25] and the populations of specialist birds are decreasing at a much higher rate than that of generalists [26]. The Three-Toed Woodpecker is specialised for high-altitude forest habitats and, as a glacial relict on the Balkan Peninsula, it is located on the edge of its range [7]. This is a cause for concern because birds at the hot edge of their range show less tolerance to global warming [26]. 

The species conservation status on the global [27] and European scope [28] is Least Concern according to the IUCN Red List category. The largest population with approximately 20,000 pairs and the largest density is found in Northern Europe, whereas South European mountains have lower densities [5]. The recent European population trend is unknown; however, the population trend within the European Union (EU) is decreasing [29]. In the EU, it is protected by The Birds Directive Annex I and Bern Convention Annex II and Revised Annex I of Resolution 6 [28]. There are 550 Natura 2000 sites designated within the EU for the conservation of the Three-Toed Woodpecker, of which 25 are found on the Balkan Peninsula: 10 in Bulgaria, 8 in Slovenia, 4 in Greece, and 3 in Croatia [28]. It was also one of the trigger species for the identification of 16 Important Bird and Biodiversity Areas in the Balkans: 5 in Serbia, 5 in Montenegro, 3 in Bulgaria, and 3 in Croatia [30]. Although the species is not globally threatened, its conservation status varies within the Balkans’ national scope. The Three-Toed Woodpecker is included in the Red Data Books and Lists of Montenegro [31], Croatia [32], and Slovenia [12] as a Near Threatened taxon, in Bosnia and Herzegovina as a Vulnerable taxon [33], in Serbia [34] and Bulgaria [35] as an Endangered taxon, and in Greece as a Data Deficient taxon [36]. The population trend in the Balkans is mostly unknown, except in Bulgaria, which has a negative trend [34]. 

To improve conservation management and ensure the species protection, distribution data are important [37]. However, as a rare [9] and one of the least studied birds in the Balkan Peninsula [7], a certain gap in existing distribution data of the Three-Toed Woodpecker can be expected, which may pose a challenge for its future conservation. In this regard, a frequently used method for defining the relationship between species occurrence and environmental determinants in various periods is species distribution modelling (SDM). It is a statistical assessment of the potential species extent and its driving factors [38]. Modelling has so far been used in several studies in Europe that have dealt with identifying the potential distribution, i.e., suitable habitats for the Three-Toed Woodpecker in the present [39,40,41] and assessing the effects of future climate changes. A narrowing of its distribution was observed in the mountain forests of Central Europe [42] and in the boreal forests of Northern Europe [43], with altitudinal and latitudinal shifts, respectively. However, to the best of our knowledge, this is the first study to assess the impacts of climate change on the distribution of the Three-Toed Woodpecker in the Balkan Peninsula. Identifying preceding changes in distribution is important for understanding potential future changes [44], which is why, in addition to the current and future periods, modelling was also performed for the past. The aims of the present study were the following: (1) to identify spatial patterns in the species distribution in the present; (2) to estimate the most important environmental variables that influence the distribution; and (3) to project the current model into past (last interglacial and last glacial maximum) and future climate conditions (2050 and 2070) for assessing the impacts of climate change on this rare and endangered species of woodpecker on the Balkan Peninsula.

## 2. Materials and Methods

### 2.1. Study Species

The Three-Toed Woodpecker is an old-growth and mature forest specialist [8]. It is characteristic of coniferous forests [9,39,41,45,46], predominantly spruce; however, it can also be found in mixed deciduous forests [47,48], especially on steep slopes [9,49]. The species’ affinity for steeper slopes is probably indirectly related to the abundance of dead, dying, or weakened trees [49], where the insects on which it feeds are present [39,47]. Diet is based primarily on the insects of the order Coleoptera, e.g., *Polygraphus* and *Ips*, but it also feeds on other insect larvae, e.g., from the Lepidoptera and Hymenoptera order, and spiders [5,34]. The larger proportion of dying trees, and therefore an increase in the availability of xylophagous insects, was shown to influence population growth [50]. Large dead and decaying trees are also important as nest cavity sites [8]. 

The Three-Toed Woodpecker is a sedentary breeding species in the Balkan Peninsula and inhabits only high mountainous coniferous and mixed forests [34]. While this species has irruptive nature in other parts of its range, there is no any research yet confirming its irruptive character in the Balkan Peninsula. The population size is estimated at 50–90 pairs in Serbia [34], less than 50–100 pairs in Bosnia and Herzegovina [33], 80–200 pairs in Bulgaria [13], 350–600 pairs in Slovenia [12], 500–1000 pairs in Croatia [11], and 1300–2600 pairs in Montenegro [51]. The main factors endangering the species at the micro- and meso-habitat level on the Balkan Peninsula are the logging of old forests [13], sanitary removal of dead and dying trees, fragmentation of habitats, and urbanisation of mountains [6]. The Three-Toed Woodpecker has been identified as a key and/or indicator species in dominantly coniferous forests first because some other bird and mammal species depend on the cavities it creates and second because it can indicate changes occurring in landscape due to its sensitivity [39,40,52,53]. As a significant consumer of xylophagous insects, it has an important role in the forest ecosystems [50], especially after natural disturbances when the number of insects can significantly increase on dead or dying trees [54]. All of this indicates the significance of the Three-Toed Woodpecker for forest ecosystems in the mountains of the Balkan Peninsula. 

### 2.2. Study Area

The Balkan Peninsula is located in Southeastern Europe, between the Adriatic and Ionian Seas in the west, the Mediterranean Sea in the south, and the Aegean, Marmara, and Black Seas in the east. However, there is still disagreement in the literature about its northern border. According to Cvijić [55], the northern border of the peninsula is located along the Danube and Sava valleys to the Ljubljana basin, which divides the Alps from the Dinarides, from where the border continues to the river Soča in the northwest. The total area of the Balkan Peninsula depends on the criteria used to delimit the northern border and is about 532,000 km^2^ according to Strid et al. [56], i.e., 667,000 km^2^ according to Telbisz et al. [57]. Based on the QGIS 3.24 area calculation [58], the continental part of the peninsula (excluding the islands) has an area of approximately 477,322.95 km^2^ (Figure 1).

The most prominent geomorphological features of the Balkan Peninsula are mountain ranges, whereby Willis [2] noted that more than 60% of its territory is above 1000 m. a.s.l. Among them, Demek et al. [59] distinguished the Dinarides (with Outer and Inner Dinarides, Pelagonides, and Hellenides), with its characteristic NW–SE direction that follows the Adriatic coast, then Rila-Rhodope massif with a W–E and N–S direction, the Stara Planina (Balkan) Mountain system (with the main ridge and the Pre-Balkan Mountains), and the East Balkan Uplands, south of the Stara Planina. A significant hypsometric difference exists, ranging from sea level (0 m) to 2925 m a.s.l. Musala Peak on Rila Mountain [57]. The river systems of the Balkan Peninsula are diverse, from mountainous rivers with steep gradients to meandering rivers of the plains. Balkan rivers contribute 26% to the European Mediterranean runoff, and the Drim and Neretva Rivers (both located in the western part of the peninsula) rank third and fourth, respectively, in the annual discharge of the Mediterranean region [60].

The northern parts of the peninsula are under the Eurasian continental influence, whereas the southern coastal parts are under the maritime influence of the Mediterranean Sea [18]. The annual temperature ranged from 9.5 °C in higher altitude regions to 17 °C along the coastal belt of the Adriatic Sea in the period 2000–2020 [61]. The climatic difference between the north and south of the peninsula is also expressed in the rainfall regime, whereby in areas with a continental type of climate, the largest amount of precipitation occurs in the period from May to August, while for a maritime type of climate, the period from October to January represents the wettest part of the year; in addition to this, there are transitional areas with two rainfall maxima during the year [18]. The highest amount of precipitation occurs in the west of the peninsula, i.e., the coast of the Adriatic and Ionian Seas, whereas the lowest amount of precipitation is recorded over the northeastern and eastern parts of the Balkans [62]. The amount of annual precipitation greater than 1000 mm per year in the west of the peninsula is conditioned by the influence of the mountains, i.e., the existence of an orographic factor in the form of the Dinarides [18]. In addition, the high mountains represent an obstacle to the penetration of warm air masses into the inland, thereby limiting the maritime influence on the coastal belt [63]. Relief is an important climatic factor in the Balkan Peninsula because of its heterogeneity, which causes pronounced variations in temperature and precipitation, often with rapid transitions and contrasts [2,63]. This is in agreement with Bennett et al. [1], who stated that the Balkan Peninsula has a steeper climate gradient than other parts of the European continent.

### 2.3. Occurrence Points

To obtain georeferenced data on the presence of species, occurrence points from the database of the Institute for Nature Conservation of Serbia [64], the online database Global Biodiversity Information Facility [65], and data collected by Marko Raković during fieldwork in the period of 2015–2024 in central, eastern, and southern parts of the Balkans (personal data) were used. Database of the Institute for Nature Conservation of Serbia contained the observation date, the number of observed individuals, the bibliographic sources, the name of the mapper, coordinates, and the species conservation status in Serbia and the world. The data was compiled from the species records in the field, literature, and from different collections, such as the Natural History Museum in Belgrade, project reports from the National Park Tara in 2014, BioRaS 2000–2012, NaturaList, and Biologer. The last three sources represent the platforms designed to collect and digitise the records on biodiversity. Biologer platform has been active in Serbia since 2018, and in the recent years Croatia and Bosnia and Herzegovina have also joined the initiative to record species observations [66]. The Global Biodiversity Information Facility is an online database that is often used in SDM research. The downloaded data for the Three-Toed Woodpecker included records from 12 datasets, of which the highest number belonged to eBird Observation Dataset. The number of points collected at the beginning of the research was 1068. Because the Three-Toed Woodpecker is present throughout the year on the Balkan Peninsula, the collected data were inspected independently of the month when the point was recorded. On the other hand, as changes in land cover are rapid and to preserve representative data, only occurrence points recorded in the period 1990–2023 were retained in the first examination. 

Occurrence points often show a spatial bias towards accessible places, such as conservation priorities, roads, rivers, and cities [67]. To reduce the effect of sampling bias, occurrence data were spatially filtered using the spThin package [68] in the R programming language 1.4.1106 [69]. Spatial filtering can help create better models that show stronger performance and broader prediction, i.e., which do not overfit the training data [70]. This method is based on a randomisation approach that aims to reduce the sampling bias by removing the fewest possible number of points, so that the important information is not lost. It employs thinning based on a nearest neighbour, which removes the occurrence records closer than a minimum distance specified by a user. The algorithm identifies the occurrence points with the largest number of neighbours and removes them until no point is closer than the specified distance [68]. When choosing the distance between the occurrence points, it was important to find a solution that would simultaneously reduce bias and allow the preservation of important data concerning the response of the species to environmental conditions. Because areas with heterogeneous relief require smaller spatial filters compared to more homogeneous terrains [70] and based on visual inspection of the data, the distance between points was set to 5 km with 10 replicates. By inspecting the obtained result in QGIS (109 points), four more points assumed to have imprecise coordinates were eliminated, bringing the total number of points for Three-Toed Woodpecker to 105 (Appendix B). The obtained number of points is not considered low for few reasons: (1) the Three-Toed Woodpecker is a habitat specialist whose presence is limited to high mountains in the Balkan Peninsula; (2) additionally, its presence is connected to forest habitats with old dead trees which are especially limited since modern forestry removes dead trees constantly; and (3) habitat specialists require fewer number of occurrence points contrary to generalist species, which are expected to be found in the wider habitats.

### 2.4. Environmental Variables

The environmental variables for the development of the SDMs were selected based on their ecological importance for the species under study, as recommended by Elith and Leathwick [71]. Environmental variables can be classified into three main categories: climatic, topographical, and land cover. All three groups were used to create the current model, whereas the past and future climate models were created only based on bioclimatic variables. This approach has been used in different studies so far [72,73,74], and it is based on the fact that the values of non-climatic factors in the model projected to the novel period remain the same as in the present, i.e., no changes can be predicted [72,74]. Pearson and Dawson [75] also stated that climate variables at the macro-spatial scale have a dominant effect on species distribution and that using them as a single variable can provide effective results that give insight into potential impacts.

Data on the current climate were obtained from the WorldClim version 2.1 database [76], which has often been used in studies dealing with SDM. WorldClim allows the download of 19 bioclimatic variables in different spatial resolutions, which are derived from monthly values of temperature and precipitation and represent averages for the period 1970–2000 [77,78]. These climatic variables are defined for a 30-year period, i.e., climate normal. As there is no 2000–2030 climate normal yet available on the WorldClim and because we wanted to see the effect of the current climate on the species distribution, we decided to use this time period. To determine which bioclimatic variables are important for the studied species, a literature review was conducted [40,42,43,79,80,81,82]. Nine variables were found to be most often used in studies dealing with boreal bird species, including the Three-Toed Woodpecker: minimum temperature of the coldest month (BIO6), mean temperature of the wettest quarter (BIO8), mean temperature of the driest quarter (BIO9), mean temperature of the warmest quarter (BIO10), mean temperature of the coldest quarter (BIO11), precipitation of the wettest quarter (BIO16), precipitation of the driest quarter (BIO17), precipitation of the warmest quarter (BIO18), and precipitation of the coldest quarter (BIO19). These variables were then retrieved from the WorldClim database at an original resolution of 30 arc seconds, which is approximately 1 × 1 km^2^.

Because bioclimatic variables are derived from the same temperature and precipitation data, they are highly correlated [83], which can lead to incorrect conclusions about their importance [84], reduce the predictive ability of a model, and increase the uncertainty of the results even with a low degree of collinearity [85]. Although certain modelling methods, such as Maxent, are less sensitive to the correlation between variables [85,86], to avoid possible issues, environmental data were first extracted from all 105 occurrence points for each bioclimatic layer separately using the Sample Raster Values function in QGIS. To solve the problem of collinearity, variance inflation factor (VIF) analysis was performed [87] using the R programming language, whereby all variables with VIF > 4 were eliminated. With this step, the number of bioclimatic variables was reduced from nine to four. As the VIF analysis offered more possible combinations of noncollinear predictors, those that make the most sense for the model from an ecological and geographical point of view were chosen, namely BIO8, BIO9, BIO10, and BIO18. Geographically, the Balkan Peninsula is not characterised by a uniform rainfall regime. Thus, for the northern part of the Balkan Peninsula, the end of spring and summer are the wettest periods of the year, whereas for the southern part of the Balkan Peninsula, that is the end of autumn and winter. The driest period, on the other hand, is summer for the south of the Balkan Peninsula, i.e., winter for its northern part [18]. Because the mean temperature of the warmest quarter was always highly correlated with the variables representing the cold period, choosing the mean temperature of the wettest quarter and the mean temperature of the driest quarter allowed the model to cover both periods—warm and cold—of the year.

In the current model, the Copernicus Global Digital Elevation Model [88] was used in the original resolution of 90 m to obtain the topographical variables of elevation, slope, and aspect. Because the land cover changes are rapid, the land cover data were obtained from the Corine Land Cover (CLC) database [89] for the most recent year 2018 in the original resolution of 100 m.

To represent the climate of the past, distribution modelling was performed in two different time periods: the last interglacial, which was approximately 120,000–140,000 years ago, and the last glacial maximum, which was approximately 22,000 years ago [90]. The same bioclimatic variables were used as in the present, whereby for last interglacial their values were obtained in an original resolution of 30 arc seconds and for the last glacial maximum in a resolution of 2.5 arc minutes. Two global climate models (GCMs), CCSM4 and MIROC-ESM, were used. All paleoclimate data were downloaded from the WorldClim database; however, as it is not yet available in the current version 2.1, version 1.4 was used [91].

For predictions of the species distribution under the impact of future climate changes, bioclimatic variables were derived from a set of four GCMs (HadGEM3-GC31-LL, IPSL-CM6A-LR, MRI-ESM2-0, MIROC6) at the original resolution of 30 arc seconds for 2041–2060 (average 2050) and 2061–2080 (average 2070) under the SSP2-4.5 and SSP5-8.5 GHG emissions scenarios. All future climate data were obtained from the WorldClim version 2.1 database. The selection of GCMs was done according to Merrifield et al. [92], by respecting the criteria that individual models do not belong to the same model families and that they differ from each other in terms of characteristics, which enables the diversity of future climate. Following Merrifield et al. [92], HadGEM3-GC31-LL was selected as a representative warm/dry model, IPSL-CM6A-LR as a representative warm/wet model, MRI-ESM2-0 as a cool/wet model, and MIROC6 as a cool/dry climate model. The use of a larger number of GCMs is recommended to avoid the problem of a single model uncertainty and to enable the observation of diversity [93]. In addition, IPSL-CM6A-LR and MRI-ESM2-0 have previously been used in a study investigating the impact of global warming on high-elevation birds [14], whereas HadGEM and MIROC have been applied in different versions to assess the climate change impact on birds [72,94,95]. All the models mentioned above are part of the current sixth phase of the Coupled Model Intercomparison Project [92], in which the previous Representative Concentration Pathways have been replaced by updated versions known as Shared Socioeconomic Pathways (SSPs) [96]. These represent potential climate futures that depend not only on socioeconomic development but also on measures to mitigate climate change [21]. SSPs 2.4-5 and 5-8.5 were chosen because they reflect intermediate and very high GHG emissions scenarios, respectively.

### 2.5. Species Distribution Modelling

Prediction of the distribution of the Three-Toed Woodpecker was performed using Maxent version 3.4.4 [97], which is a frequently used method in SDM research. For Maxent parameter settings, cross-validation with 10 replicates was selected as the replication method. This is a technique that divides the occurrence points into *k* subsets (in this case 10) to use for training and testing of the model [98]. We chose to create response curves, which allow us to understand how habitat suitability for the species changes depending on the change in the environmental variables value. In addition, the settings option to calculate the jackknife test was selected, which enables the identification of variable importance for the model based on their individual impact [86]. The relative contribution of the environmental variables changes depending on the metric for assessment because of which, e.g., the order of variables by importance between the methods may differ. The jackknife test evaluates the importance of each environmental variable by creating a model with only one variable, and then creating a model without the same variable [99]. The value of the percent contribution depends on the path that Maxent uses when creating the model; on the other hand, the value of permutation importance depends on the final model, where the contribution of each variable depends on the random selection of training points [99]. As the output format for estimating the values of habitat suitability in each raster pixel, we chose a logistic format with values ranging from 0 to 1. Furthermore, to transform the continuous values of the logistic output into a binary map, the maximum test sensitivity plus specificity threshold was applied, which minimises two types of errors: commission (false positive) and omission (false negative) [100]. Other Maxent parameter settings were kept as the default following the results of Phillips and Dudík [101], who highlighted that default settings achieve good performance, whereby regularisation is 1, background points 10,000, and the maximum number of iterations (runs) 500. Maxent was allowed to choose the most optimal set of features based on the number of occurrence points [102] because this enables more accurate modelling of the complex response of a species to environmental conditions [99]. Features represent the mathematical transformations of variables [98], and the shape of the obtained response curves depends on their choice [99].

All data were previously resampled to the same spatial extent (1 × 1 km^2^) and dimensions using the QGIS software (v 3.24) and then converted to the appropriate format necessary to work with Maxent (.csv for occurrence points and .asc for environmental variables). First, a model of the current distribution was generated, which was then projected onto the past and future using the corresponding bioclimatic variables. As a final result, 20 SDMs were obtained, of which one model represents the present, three models represent the past (last interglacial and two last glacial maximum), and 16 models future conditions (two years and two scenarios according to four GCMs). To evaluate model performance, a threshold-independent measure [103] receiver operating characteristic area under the curve (AUC) was used, which indicates the ability of the model to distinguish between the presence locations and background points [98]. As Maxent works with presence-only data, background points can be used as pseudo-absences to calculate AUC values [101]. AUC < 0.5 indicates performance that is worse than random, AUC of 0.5 random prediction [104], while for interpretation of results AUC > 0.5, it is recommended that values between 0.50 and 0.60 represent failed model, AUC between 0.60 and 0.70 is poor model, models with AUC 0.70–0.80 are fair, AUC between 0.80 and 0.90 implies good model, while AUC values 0.90–1 indicate an excellent model [105].

The models produced through Maxent were then imported into QGIS, where the continuous values of the logistic output were reclassified into suitable/unsuitable habitat maps based on the obtained threshold values. To standardise the threshold value and compare different models, we calculated its average value, which for all 20 models is 0.25758. This implies that anything below the specified threshold indicates an unsuitable habitat, whereas anything above this value can be considered a suitable habitat [99]. Following Cerman et al. [106] and Zhu et al. [107], suitable habitats were further divided into three categories: poorly suitable habitat (0.25758–0.65758), moderately suitable habitat (0.65758–0.85758), and highly suitable habitat (0.85758–1).

## 3. Results

### 3.1. Maxent Statistics

The results of the jackknife test for past and future models are identical; therefore, mean temperature of the warmest quarter, mean temperature of the driest quarter, and precipitation of the warmest quarter were identified as the three most important environmental variables for creating the model, respectively, whereas mean temperature of the wettest quarter in all 19 models was in the fourth and last place in terms of contribution. According to this jackknife test, the environmental variable that contributes most to the creation of the SDM in the current period is still mean temperature of the warmest quarter, followed by mean temperature of the driest quarter in second place, with the novelty that elevation and CLC are ranked third and fourth in importance, respectively. The fifth most important environmental variable in the current period is precipitation of the warmest quarter, whereas mean temperature of the wettest quarter, slope, and aspect are the three variables with the least contribution to the model. According to the percent contribution, the five most important environmental variables in the current model are mean temperature of the warmest quarter (31.5%), CLC (26.2%), elevation (21.8%), precipitation of the warmest quarter (16.6%), and mean temperature of the driest quarter (2.2%), whereas according to the permutation importance after mean temperature of the warmest quarter (56.3%), the second most important environmental variable is precipitation of the warmest quarter (18.6%), followed by CLC (11.5%), mean temperature of the driest quarter (5.1%), and mean temperature of the wettest quarter (3.8%) (Appendix A). By evaluating the modelling results, it was found that the Test AUC value for all 20 models was >0.90.

With the increase in altitude, habitat suitability increases up to 1801 m a.s.l. This positive correlation was also observed with the slope, where the habitat suitability increases up to 38°. The third topographic variable was aspect, and the results indicate that the north-facing slopes are the most optimal for the studied species. The three CLC classes whose presence contributes the most to the potential presence of the species on the Balkan Peninsula are sport and leisure facilities, coniferous forests, and mixed forests. The mean temperature of the wettest quarter indicates that the optimal habitat for the species is associated with low temperatures (>−5 °C < 5 °C), whereby the greatest suitability corresponds to a mean temperature of 1.6 °C. The mean temperature of the driest quarter records the peak of habitat suitability at −5 °C, but another suitable temperature range (>5 °C < 15 °C) is observed, with the most suitable habitat at 11 °C. Habitat suitability for the mean temperature of the warmest quarter reaches its peak at 11 °C. With an increase in the amount of precipitation of the warmest quarter, the habitat suitability increases until 472 mm (Appendix A).

### 3.2. Species Distribution

In the current period, the total area of suitable habitats for the Three-Toed Woodpecker on the Balkan Peninsula is 31,849 km^2^, of which the largest area is occupied by poorly suitable habitats (26,217 km^2^), followed by moderately suitable habitats (5257 km^2^), while highly suitable habitats, with an area of 375 km^2^, have the smallest share (Appendix A). The rest of the territory of the Balkan Peninsula (93.33%) consists of habitats unsuitable for the studied species (maximum test sensitivity plus specificity threshold < 0.25758). Concerning the spatial distribution, the highest percentage of highly suitable habitats is located on the territory of Serbia (0.36% of the total area of the country), followed by Croatia (0.30%), Montenegro (0.22%), and Bosnia and Herzegovina (0.13%), while 0.02% and 0.01% are found on the territory of Bulgaria and the Balkan part of Slovenia, respectively. The distribution of moderately suitable habitats ranges from 3.69% in the territory of Montenegro, followed by Bosnia and Herzegovina with 2.72%, Croatia with 2.17%, part of Slovenia with 2.13%, and Serbia with 1.92% of its total territory. Bulgaria, Albania, and North Macedonia have a share of moderately suitable habitats < 1%. The territory of Montenegro recorded the highest percentage of poorly suitable habitats (21.61%), followed by Bosnia and Herzegovina (14.98%), Serbia (8.65%), part of Slovenia (8.59%), Croatia (5.63%), Bulgaria (5.02%), and Albania (1.50%). North Macedonia and Greece participate with <1% of poorly suitable habitats.

The percentage of change in the total area of suitable habitats between the present and the last interglacial is −99.19%. It is predicted that 247 km^2^ of the area consisted of poorly suitable habitats and 10 km^2^ of moderately suitable habitats, whereas there were no highly suitable habitats for the Three-Toed Woodpecker on the Balkan Peninsula 120,000–140,000 years ago (−100% change). On the other hand, the average total area of suitable habitats for the two last glacial maximum models is 179,325.51 km^2^, which is a 463.05% larger area than that in the current period. In all three categories, the suitable area on the Balkan Peninsula was larger than that today, but the greatest change was that highly suitable habitats occupied an average area of 36,211.01 km^2^ (9556.27% more). In contrast to the present, where highly suitable habitats have the smallest share in the total area of suitable habitats (1.18%), this category was in the second place (20.19%) during the last glacial maximum, right behind poorly suitable habitats, which were also dominant in that period (61.59%), although less than today (82.32%). Comparing the two last glacial maximum models, MIROC-ESM showed a larger total area of suitable habitats (207,062.01 km^2^) than CCSM4 (151,589 km^2^). Considering the ecology of the species, i.e., its affinity for cooler temperatures, these results indicate that the first model shows a slightly colder climate in the given period of the past than the second model (Figure 2).

In contrast, all 16 future models showed a decrease in the total area of suitable habitats compared to the current period, with the average value for SSP2-4.5 (moderate scenario) being −75.89% in 2050 and −83.86% in 2070, while SSP5-8.5 (the worst scenario) shows −86.62% of change for 2050 and −96.38% of change for 2070. All three categories of suitability show a negative sign (Appendix A). 

Although all future models showed a decrease in habitat suitability on the Balkan Peninsula, there are still certain differences between them. The GCM that showed the mildest scenario compared with the other models was MRI-ESM2-0. This model according to Merrifield et al. [92] belongs to the group of cool/wet climate models, and the obtained results indicate a total area of suitable habitats of 12,141 km^2^ for the moderate scenario in 2050, which is also the highest value of suitable area among all future models. In addition, this is the only climate model that showed the presence of highly suitable habitats in all future years and scenarios. On the other hand, the GCM that showed the greatest warming is HadGEM3-GC31-LL, which belongs to the group of warm/dry climate models according to Merrifield et al. [92]. It has the lowest value of the total suitable habitats among all 16 models (54 km^2^ for the worst scenario in 2070). According to the same model, highly suitable habitats for the species are present only in the moderate scenario of 2070, with an area of only 1 km^2^. On average, the difference in total area of suitable habitats between the mildest model (MRI-ESM2-0) and the most extreme model (HadGEM3-GC31-LL) is 5815 km^2^ in the moderate scenario and 4792 km^2^ in the worst-case scenario. Among the other two models, MIROC6 (cool/dry model according to Merrifield et al. [92]) compared with IPSL-CM6A-LR (warm/wet model according to Merrifield et al. [92]) showed a larger total area of suitable habitats for all years and scenarios, except in the moderate scenario of 2070, with the difference of 440 km^2^ between the models. On average, the difference in total area of suitable habitats between MIROC6 and IPSL-CM6A-LR is 1572 km^2^ in moderate scenario and 813 km^2^ in the worst-case scenario. MIROC6 showed a lack of highly suitable habitats for the species in 2070 for both GHG emissions scenarios, unlike IPSL-CM6A-LR, which showed this lack only in the worst-case scenario of 2070.

Based on moderate scenario, the largest average difference in total area of suitable habitats is between the cool model and the warm models, i.e., mildest model MRI-ESM2-0 and IPSL-CM6A-LR (6093 km^2^), and the mildest model and the most extreme model HadGEM3-GC31-LL (5815 km^2^). The smallest difference is between the two warm models, HadGEM3-GC31-LL and IPSL-CM6A-LR, 1376 km^2^ (Figure 3).

Based on the worst-case scenario, the largest average difference in total area of suitable habitats is also between the cool model and the warm models, i.e., the mildest model MRI-ESM2-0 and the most extreme model HadGEM3-GC31-LL (4792 km^2^), and the mildest model and IPSL-CM6A-LR (4682.5 km^2^). The smallest difference is between the two warm models, HadGEM3-GC31-LL and IPSL-CM6A-LR, 109.5 km^2^ (Figure 4).

As in the models for the present and the past, the largest share of the total area of suitable habitats in all future models goes to poorly suitable habitats. According to the moderate scenario, this category accounts for an average of 93.59% of the area in 2050 and 93.34% of the area in 2070. Of the remaining total area, an average of 5.95% belongs to moderately suitable habitats in 2050 and 6.13% in 2070, whereas highly suitable habitats participate with an average of 0.46% in 2050 and 0.53% in 2070. Under the worst scenario, there is an increase in the representation of poorly suitable habitats with an average of 94.97% of the area in 2050 and 98.55% of the area in 2070. Their expansion can also be interpreted as a decrease in the area of the other two categories in favour of poorly suitable habitats, as shown by the results. According to the SSP5-8.5 scenario, moderately suitable habitats are represented by an average of 4.55% in 2050 and by 1.34% in 2070. Highly suitable habitats have the smallest share, with an average of 0.48% in 2050 and 0.11% in 2070.

## 4. Discussion

### 4.1. Environmental Determinants

With an increase in altitude, habitat suitability for the species also increases, but only to a certain extent. The decrease of habitat suitability above 1801 m a.s.l. can be explained by the height of the tree line in the study area. The transition between the forest zone and the high mountain vegetation zone varies among the mountains of the Balkan Peninsula and amounts to, e.g., between 1700 and 2200 m in the central part of the peninsula and 1800 and 2000 (2300) m in the south of the peninsula, while on the eastern and southeastern mountains of the peninsula it is between 1900 and 2500 m [56]. Bearing in mind the characteristics of the studied species, it is expected that high altitudes due to the lack of forest, i.e., the presence of subalpine and alpine vegetation, are unsuitable. The model also shows the species affinity for steeper slopes and this result is in agreement with the literature [9,49]. Coniferous and mixed forests belong to the classes that contribute the most to the potential presence of the species on the Balkan Peninsula, which is consistent with the known ecology of the species. The novelty shown by the current model is the dominance of sport and leisure facilities, followed by the two forest classes. According to the CLC nomenclature [108], this class includes areas intended for sports and recreation, campsites, ski resorts that use artificial snow, visitor centres of protected areas, resorts, cottage areas, and more. However, the presence of the Three-Toed Woodpecker is usually negatively correlated with built-up areas because it is sensitive to land conversion [39,54]. All these findings indicate an increased human pressure on the environment that the Three-Toed Woodpecker probably inhabits. In addition, the value of the logistic output for sport and leisure facilities and coniferous forests is almost identical (0.78 and 0.74, respectively) (Appendix A), which indicates a possible spatial connection of the mentioned artificial surfaces and forests and could represent a potential threat. In all time periods, the mean temperature of the warmest quarter was identified as the most important environmental variable, i.e., the variable that in itself has the most useful information for estimating the species distribution. This result is not surprising because the Three-Toed Woodpecker is a boreal bird species that is negatively affected by high temperatures. This is confirmed by the response curve for the given bioclimatic variable, where the parts of the area with a mean temperature ≥ 20 °C are unsuitable for the studied species, whereas the most suitable habitat is provided by a mean temperature of 11 °C. The affinity towards lower temperatures is also shown through the other two response curves related to the mean temperature of the wettest quarter and the mean temperature of the driest quarter. Taking into account the given values for all three temperature variables, it can be concluded that a mean air temperature below approximately 15 °C enables the survival of the species on the Balkan Peninsula, i.e., this value is potentially considered as the upper limit (maximum).

### 4.2. Species Distribution

Analysis of the current distribution shows that the largest part of the Balkan Peninsula is unsuitable for the species, whereas suitable habitats make up only 6.67% of the peninsula’s territory. These data indicate a narrowly specialised bird species, as opposed to generalists, which are more widespread due to their greater tolerance to different environmental conditions. Therefore, the parts of the territory that are characterised by highly suitable environmental conditions for its presence and survival are of essential importance for the Three-Toed Woodpecker. However, these areas currently occupy the smallest part of the total suitable area, with no Balkan country having a share of highly suitable habitats above 1% on its territory. On the territory of Albania, North Macedonia, Greece, and the Balkan part of Turkey, highly suitable habitats are not present. It is important to keep in mind that a possible explanation for their lack in the territory of these four countries is the application of a high threshold for the category of highly suitable habitats (>0.85758). This was done in order to identify areas with the most optimal conditions for a specialist species. The territory of Turkey did not show the presence of favourable conditions in any of the three categories, meaning that the Balkan part of its territory is characterised by an unsuitable environment for the presence of the species. This result is in agreement with the known data on the countries where this species is present [30]. The other Balkan countries participate in at least one (Greece), two (North Macedonia and Albania), or all three categories of suitability (Montenegro, Bosnia and Herzegovina, Serbia, Croatia, Slovenia, and Bulgaria). Suitable habitats have the largest presence on the territory of Montenegro, which supports Rubinić et al. [51] that more than 50% of the Three-Toed Woodpecker population in the Balkans is present in that country. Other than that, Bosnia and Herzegovina, Serbia, and Croatia alternate as the first three countries with the largest presence of suitability. Therefore, it can be concluded that the western part of the peninsula records a higher percentage of favourable environmental conditions than the southern and eastern parts.

The time periods, based on the size of the area suitable for the Three-Toed Woodpecker, are presented here starting from the largest to the period with the smallest area: last glacial maximum > present > future > last interglacial. The modelling results indicate that the Balkan Peninsula was characterised by the most favourable environmental conditions for the survival of the species around 22,000 years ago, compared with the other analysed periods. During the glaciation, the temperature in Southern Europe was 8–11 °C cooler than today [109], with the July temperature in the Balkans being 5 °C lower [2], whereas the January temperature was about −7 °C to 3 °C [1]. Such low values probably contributed to the species inhabiting a wider territory of the Balkans than today. The larger presence of highly suitable habitats is an important indicator of greater suitability of the peninsula for this specialist species during last glacial maximum. At the same time, this signifies that after the temperature started to rise at the end of the last glacial maximum, the species distribution began to contract to the area where somewhat colder conditions persisted. In the case of the Balkans, these parts of the territory are today represented by mountains. However, it cannot be said that the analysed past has been entirely favourable for the Three-Toed Woodpecker. During the last interglacial, which was one of the warmest periods in the recent geological past [21], the temperature was higher than today by approximately 5 °C [110]. Hence, it is not surprising that this period was less favourable for the species, with only 257 km^2^ of the total area of suitable habitats. It can be observed that climate oscillations during the geological past led to a change in this species extent, whereby during colder periods conditions for an expansion would be created, which would then contract during warming. Based on this, it is expected that changes (a decrease) in the distribution of the Three-Toed Woodpecker will continue in the future, with further climate changes, which models of future climate show.

Projected global warming in both climate scenarios led to a decrease in the areas with suitable conditions. The potential species distribution changes as follows: average 2050 SSP2-4.5 > average 2070 SSP2-4.5 > average 2050 SSP5-8.5 > average 2070 SSP5-8.5. The moderate scenario shows a slightly larger suitable area (7678.75 and 5141.75 km^2^, respectively) than the worst scenario (4261.5 and 1154.5 km^2^, respectively), and within each, habitat suitability decreases with approaching 2100. The drastic reduction in highly suitable cells is a cause for concern. Regardless of the GHG emissions scenario, the Three-Toed Woodpecker is predicted to lose an average of 90% of highly suitable habitats. This number ranges from −90.53% in 2050 (moderate scenario) to as much as −99.67% in 2070 (worst scenario). This would mean that the parts of the Balkan Peninsula that are most optimal for its survival would almost completely disappear. As birds are highly mobile species, it can be assumed that because of the described changes, the Three-Toed Woodpecker will probably be forced to inhabit the next category of suitability, if conditions for that exist. However, even for moderately suitable habitats, the predictions of area reduction are not milder than those for highly suitable habitats, i.e., the average loss is above 90% in all scenarios and years. This represents a threat to future conservation and implies that the only remaining category of suitable habitats would be areas of poor suitability. Although there is also an evident area decrease, it is still somewhat smaller than that in the previous two categories, and only under the worst-case scenario of 2070 is the average area reduction larger than 90%. These results confirm that climate change represents a threat of high importance to the Three-Toed Woodpecker [111]. Compared with Bollmann et al. [42], who also found a narrowing of its distribution of −22% in Central European mountains in 2050, the decrease of suitable habitats in the Balkan Peninsula is much larger in extent. Additionally, the Three-Toed Woodpecker was the least susceptible to climate change effects compared with other analysed cold-adapted birds in the Central Alps, Northern Prealps, Swiss Jura, and Black Forest [42]. These differences point out the need for regional SDM assessments and signal that the Balkan Peninsula is facing enhanced warming, causing a threat to all boreal plant and animal species there.

### 4.3. Conservation Recommendations

Our results can be crucial for the conservation because they indicate that the human role would potentially have to be greater for the species to survive in the Balkan Peninsula. This is important for all three categories of suitability: primarily for highly and moderately suitable habitats, which must be a conservation priority, but also for the category of poorly suitable habitats, which, according to SDMs, will have the smallest percentage of change compared with the present. In the case of the third category, as it represents an area where it is not certain that a narrowly specialised species could successfully survive without any help, conservation activities must be of greater extent. One way to mitigate the effects of climate change on the species is to address existing conservation problems at the meso- and micro-levels. The Three-Toed Woodpecker prefers naturally occurring dead and dying trees; however, modern forestry, i.e., the removal of such trees, endangers it [6,47,53] and puts additional pressure on the survival of the species that already exists due to climate change, as the modelling results show. In agreement with this is Matvejev (as cited in [13]), who indicated that the warming of the climate, together with the logging of old forests, is the main reason for the disappearance of this species from some parts of the Balkans during the 20th century. 

Recommendations for the conservation of the Three-Toed Woodpecker in the study area include, but are not limited to the following: (1) creating a buffer zone in forests, with a special focus on forests within protected areas; (2) limiting the sanitary removal of trees associated with its presence, i.e., spruce [11,45,47,48], fir [7,9,11,52], pine [7,45,47,52], birch [47], and beech [9,11]; (3) extending the time between logging [112]; (4) preserving standing dead wood in amount that is not smaller than 20 m^3^/ha [34]; (5) preserving mature and old-growth forest stands by leaving minimum 15 spruce and fir trees/ha with a diameter no less than 40 cm [34]; (6) creating a larger area of suitable habitats as the species does not favour small forests [46]; (7) banning human activities in the breeding period (March–July) in a 500 m radius from the known nest sites [34]; (8) banning the land-use changes in the species habitat, including the new forest roads construction [34]; (9) establishing connection between habitats through corridors [112]; (10) increasing the protection of areas with suitable habitats. Habitat protection and limiting sanitary logging has so far proved to be an adequate measure for the conservation of the Three-Toed Woodpecker in Bulgaria [13,45]. Where it is not possible to fully protect the area, it is necessary to introduce a special management regime, as recommended by Stachura-Skierczyńska et al. [41]. Special attention should be paid to those parts of the Balkan Peninsula where distribution modelling has identified the largest proportion of suitable habitats for the survival of the species, which is the territory of the Western Balkans, i.e., the mountain belt of the Dinarides. The geographical position of the Dinarides along with topographical heterogeneity probably provides suitable environmental conditions for the survival of the Three-Toed Woodpecker, not only because of lower temperatures but also because of favourable conditions for the development of coniferous forests, which are positively affected by humidity. As a consequence, the greatest number of conifer species in the central part of the peninsula is recorded within this mountain system [113]. Practising sustainable forestry activities such as partial cutting and maintaining natural disturbance regimes [114], i.e., creating supplies of dead and dying trees [53], can be an important step for the maintenance of this species in all three types of suitable habitats, not only in the present but also under the influence of future warming.

## 5. Conclusions

The results of this research indicate that climate warming has a negative effect on the distribution of the Three-Toed Woodpecker, which had the greatest expansion on the Balkan Peninsula during the last glacial maximum. Afterward, this cold-adapted bird species responded to the warming by narrowing its distribution to areas where favourable conditions remained, which is why it is currently limited to only high altitudes. Today, the largest territory of the Balkan Peninsula contains unsuitable environmental conditions for the species to inhabit, while highly and moderately suitable habitats have a patchy distribution, which can make its conservation difficult. Special attention should be paid to the western part of the Balkans, where the highest percentage of all three categories of suitable habitats was recorded. However, the studies for the Western Balkans [16,115] showed that the increase in the average annual temperature and the decrease in the average annual precipitation will continue in the future. Although altitudinal movement is one of the species’ responses to warming, the question arises as to how long will it be able to move vertically due to the decrease in the availability of altitudinal space with suitable conditions? In addition, it is important to keep in mind the limitations of the models. The models are based on the known data about the species presence which can be biased towards sampling. They do not take into account a variety of factors that can also affect the species distribution, such as environmental degradation, rapid land-use changes (e.g., transformation of land into urban area), habitat destruction by humans. When it comes to future climate models, for which only bioclimatic variables were used, it is important to acknowledge that their results can only be explained as an assessment of the climate change impacts. However, it is evident that the impact of non-climatic factors—primarily land-use changes and habitat destruction by humans—will continue in the future, as well. A large part of the Balkan Peninsula has already been changed under anthropogenic influences, e.g., many forest areas have been replaced by open fields [2]. The synergy of forest degradation and devastation with climate change, which according to Quante [116] can no longer be avoided regardless of the emissions scenario, requires conservation improvement. As the negative consequences of climate change can be reduced through adaptive management [117], it is necessary to ensure not only the quality of suitable habitats but also better connectivity between them [118], which would allow easier adaptation of the Three-Toed Woodpecker to expected changes. The obtained results on the range of possible consequences of climate change and the proposed conservation measures can represent a starting point for further research and regional planning, to preserve this glacial relict on the Balkan Peninsula.

## Figures and Tables

**Figure 1 animals-14-01879-f001:**
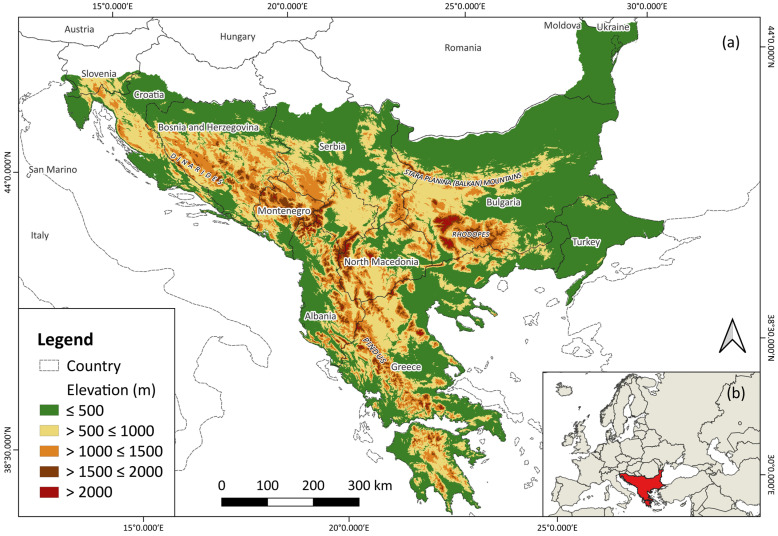
(**a**) Study area (continental part of the Balkan Peninsula; (**b**) position of the Balkan Peninsula in Europe. The northern border of the peninsula is based on Cvijić [55].

**Figure 2 animals-14-01879-f002:**
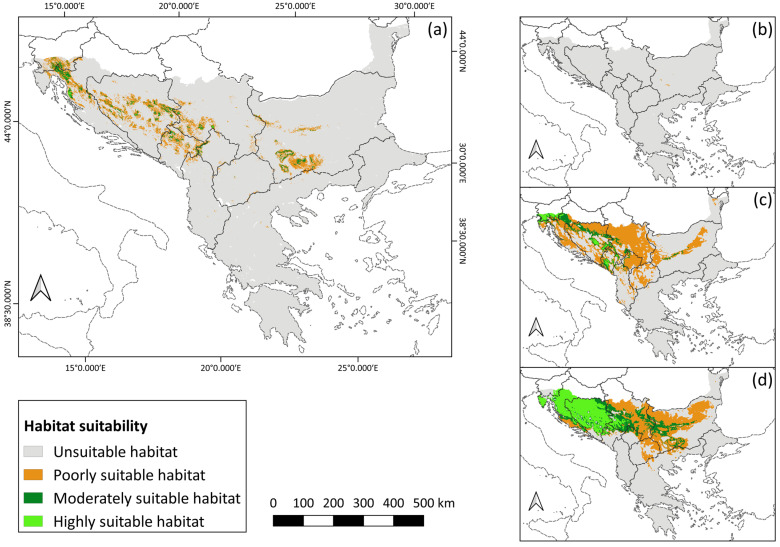
Habitat suitability maps of (**a**) the present, (**b**) last interglacial, (**c**) last glacial maximum CCSM4, and (**d**) last glacial maximum MIROC-ESM.

**Figure 3 animals-14-01879-f003:**
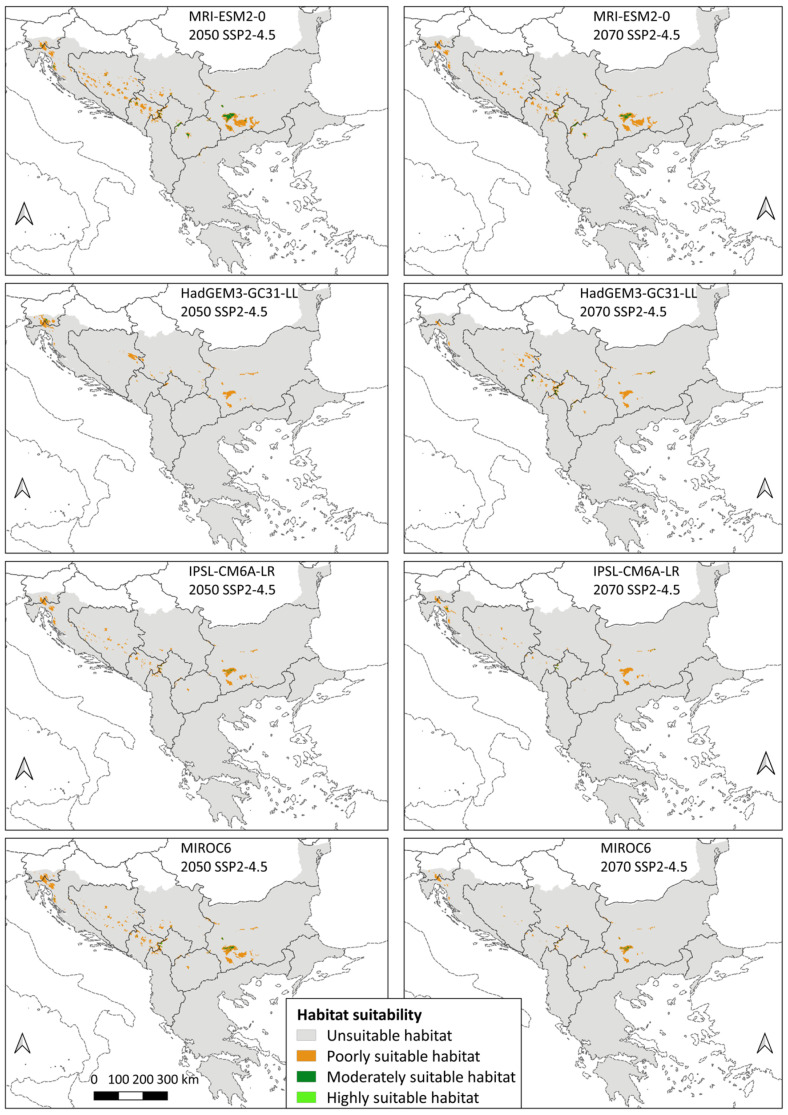
Habitat suitability maps of future conditions based on moderate scenario.

**Figure 4 animals-14-01879-f004:**
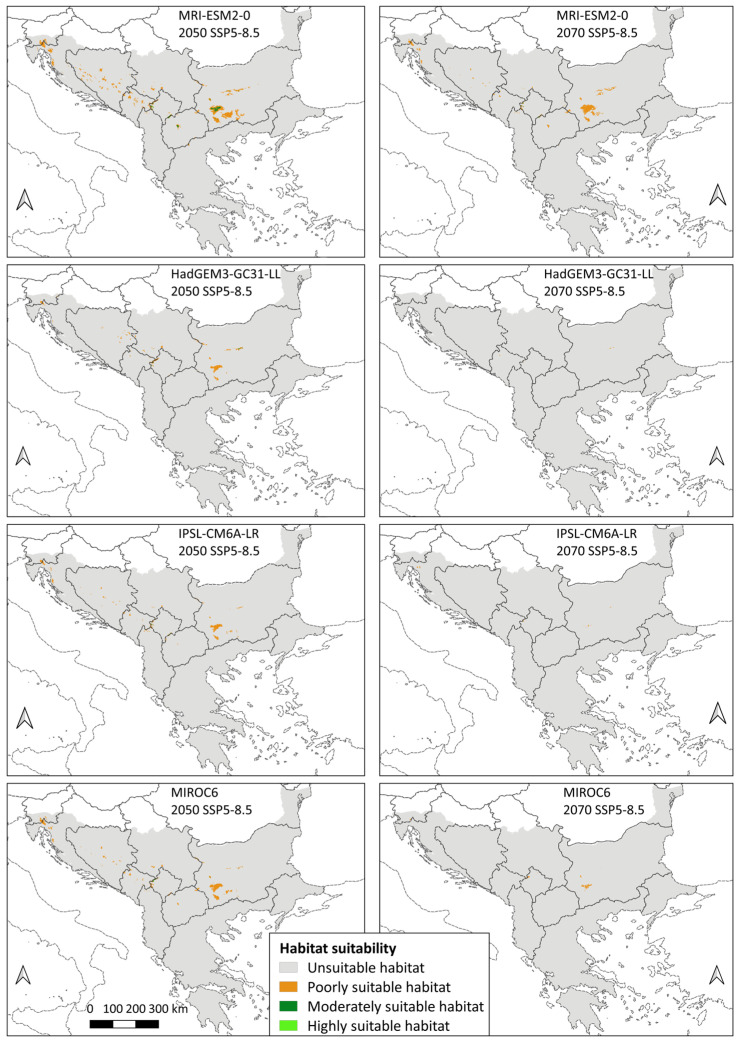
Habitat suitability maps of future conditions based on the worst-case scenario.

## Data Availability

Georeferenced data on the presence of species are publicly available in the online database Global Biodiversity Information Facility at [65]. The data obtained from the database of the Institute for Nature Conservation of Serbia were given to T.P. only for the purposes of her PhD thesis. Requests to access this dataset should be directed to the Institute for Nature Conservation of Serbia. Climate data, Copernicus Global Digital Elevation Model, and Corine Land Cover are publicly available from the following resources: [78,88,89,90], respectively.

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
