# Peer review of "An Assessment of the Climate Change Impacts on the Distribution of the Glacial Relict Woodpecker Three-Toed Woodpecker Picoides tridactylus"

_animals, 2024, doi:10.3390/ani14131879_

Round 1

Reviewer 1 Report

Comments and Suggestions for Authors

The study aims to identify the current distribution of the Three-toed Woodpecker and project its distribution under past and future climate conditions. The findings indicate that warming will negatively impact the species, reducing suitable habitats. Conservation efforts should focus on preserving areas in the western Balkans with the highest percentage of suitable habitats. This research and paper have merit and can provide useful information supporting species conservation. The primary issues with the paper are the structure and presentation of the results and discussion. Much editing and refocus is necessary to improve the impact of the paper. In addition, key limitations and issues related to the presence only data, the irruptive population nature of the species, and the modeling of a region that, by its very nature is considered marginal habitat. Please see specific comments below for context.

57: Can you please clarify the statement “owing to the possibility of altitudinal movement”?

70: The authors mention it is a rare species, with isolated populations, and the current habitat appears to be a relic of the last ice age. Does this mean the species is declining in the region naturally (regardless of contemporary human induced climate change? It seems from line 132, this is unknown

92: 40% compared to what historical period?

93: Consider moving this sentence to after the sentence that runs from 81-83. Then explain how much of the change in temp and precip is natural variation and how much is enhanced by climate change? It’s not clear if the post mid 20th century climate is natural variation or climate change induced.

216: As part of the analysis, did the authors evaluate the range of P. tridactylus (i.e., outside the Balkans) climatic niche and determine how representative the 105 occurrence points (line 230) are for the species? This could inform the plasticity of the species and possibly inform how close to the climatic niche margins the species exist within this region.

221: How does spatial filtering accomplish this? A brief explanation would benefit this section.

223: Choosing what distance? Something seems missing here.

240: What exactly do the authors mean by this? One could imagine significant and “permanent” changes to the landscape ( e.g., conversion of vegetation land cover to anthropogenic development) would be a “main driver of drastic change”. If the authors do not have a compelling reason to retain this statement and elaborate on it, please consider removing it.  

284: To clarify, the climatic niche of the species was determined based on 105 points in the Balkan’s region, which represents the margin of the species climatic niche, and this potentially narrow and marginal part of the climatic niche was used to project the distribution of the species within the LIG and LGM. Doesn’t this mean that the hindcasting  SDM represents the potential distribution at the margins during those two paleoclimatic periods? If so, what is informative about this? And if the species occurrence within the Balkans is a relic of the last glacial maximum, and there is a potential that contemporary pre-human induced climate change climatic conditions do not confer climatic niche conditions that are optimal or improve fitness. i.e., if this species is a relic, how informative is the climatic niche being used as representative.

371: The authors should consider starting the results section with the major outcomes of the SDM results and report the predictive ability of the models secondarily. i.e., the AUC results are not the interesting findings of the research, they are supportive information.

381: Why wouldn’t they be identical? They are both based on the same predictive model. Can you explain why this is interesting or under what conditions you would expect them not to be identical?

382: The use of the climatic variable shorthand e.g., BIO10, BIO9…, is convenient and saves space, but for the reader cross-tracking the shorthand to the variable is less than ideal. Please consider using the full variable name, or creating a table for quick cross reference, instead of the current listing in the Methods text (line 250-256)1

393: Can the authors please clarify this sentence, the statement “that the value changes” is not clear.

393-408: The authors should consider presenting this information in a different format. How much of this is already presented in Table 1? Can this be relegated to an Appendix? Can this be organized in a table? What key points can be highlighted in the Results section without reporting all the details directly in the Results narrative?

Table 1: The authors should consider redesigning this with the climatic variables as columns and the %s as cell values cross-tabbed with the climate model rows. Also, this table is probably more fitting for an Appendix. Like the above comment, just highlight the key variables or takeaways for the Results section. It is not readily apparent.

423-429: This is indicative of the overall need to rewrite the results section. This is presented more concisely in Figure 2 without a need to use the narrative to describe the function and shape of the curve related to the dependent and independent variables. The most interesting result is buried in line 428 “the results indicate…”. Please consider redoing the entire results section with a more concise approach and focus the narrative on the important results while relegating a lot of the support information to an appendix, table, and/or figure.

Figure 3: it is not apparent why the two future climate scenario models were used

Figure 4, 5 and lines 503 – 561: There is too much detailed information, not enough concise key result focus, and not enough key information to provide any level of interesting difference between any of the climate models. Yes, it is clear the results are different among all 16 future climate models, but what is the real degree or range of difference, and what makes the outcomes important? It is not clear from the this section.

563-573: How much is this is an interesting finding? This was expected a priori because the climatic variables were intentionally selected for their presumed fit. This is not a novel outcome, just confirmation the correct variables were selected, nothing new was learned here, thus this does not belong as the lead statement in the Discussion section.

598: Areas with anthropogenic land use had a higher probability of presence? How did you come  to the conclusion at line 607? Could this be a function of the Presence only data, i.e., landscapes more likely used by humans, are also more likely to be location where humans detect the species and not because the species is more likely to use those locations?

662: This sentence is not clear

706-707: All human disturbance or land conversion?

709: If the intention is for this relationship to be made clear and obvious in this paper, it is not.

725, 741, 697: the scientific name had been used throughout the paper, and the genus shorthand P. has been mostly used throughout the paper

754: This species is also an irruptive species, which is not addressed at all in this paper, especially with regard to the methods and use of presence-only data, which can have a huge limitation on the development of predictive models related to woodpeckers. Also, a discussion about the limitations of the models and approach is not presented and should be.

Comments on the Quality of English Language

The writing can be made more concise, and in some places, editing is necessary to improve understanding of the narrative. 

Author Response

Dear reviewer,

Reviewer 2 Report

Comments and Suggestions for Authors

The manuscript deals with a very important topic and I have no doubt that the results of the work will be applied in practice and will help in the effective protection of the Three-toed Woodpecker. I very much appreciate the enormous amount of work that the authors have put into the preparation of the manuscript. The paper may be of interest to a wide range of readers, but I think that its length and the large amount of detailed information, especially in the Material and Methods and Results sections, may make it difficult to understand (which I think the authors are aware of). Admittedly, this does not detract from the value of the paper.

In my opinion, the Introduction is a little too long and too detailed, but this does not pose any major problem in reading the manuscript.

Line 60-62, This is unnecessary information

Line 112-133 I think the information on the conservation status of the Three-toed Woodpecker is too detailed. I would suggest, for example, omitting the listing of the various documents on the conservation of this species.

The Results section is quite long and I must admit that it is quite difficult to read. A person who is not closely familiar with the work's subject matter and methods must constantly look through the Methods section to see, for example, what the various abbreviations mean. I understand that this is the nature of this type of work and it is difficult to present the results in a more readable way. May be some of the results could be presented as a supplement. This is just a suggestion.

Although the Discussion provides guidance on what actions are conducive to the conservation of the Three-toed Woodpecker, a paragraph with a number of specific recommendations based on the present study could be added.

In conclusion, the work is of great value, although its length and the enormity of the results presented may make it difficult to be received by a wider readership.

Author Response

Dear reviewer,

Reviewer 3 Report

Comments and Suggestions for Authors

I was pleased to have the opportunity to review the manuscript entitled "An assessment of the climate change impacts on the distribution of the glacial relict woodpecker Picoides tridactylus". This paper aims to assess the past, present, and future distributions of a specialist species in the Balkan Peninsula, mostly associated to conifer forests. The main results are most of the Balkan Peninsula has an unsuitable environment for the species, and all future models show a decrease in the area of suitable environment compared with the current period, suggesting hat global warming has a negative effect on the distribution of the species.

I believe this paper has interest to the scientific community, but presents some flaws that need to be fixed:

- The manuscript is very too long, and there are too many details presented that need to be cut, especially in the Methods and Results sections. The authors need to retain only the information that is really needed to understand their methods and results but not explain everything. For instance, the firsts paragraph about SDM (lines 315-325), just saying you used Maxent version 3.4.4. is sufficient here, all the rest is not needed, and you can continue with the following paragraph. Another example, figures and tables present a lot of information, and you do not have to write all of them in the Results section. For instance, all the sentences from lines 393 to 447 can be synthesized by max. 10 lines of text and in referring to Table 1 and Figure 2.

- The amount of ATTW observations is quite low (317) for the very large extent of the study area (many hundreds of km2), and it's not clear from which regions/countries they come from. Hence, a large part of the modelling should not be covered by sampling, being then extrapolation

- Furthermore, how the authors can justify the use of ATTW observations coming from a large period of time (1990-2023) with climate data averaging the conditions for the period of 1970-2000 (only 10y of overlap between both dataset), and with CLC habitat classification which is for the year 2018 (so, going to -28y up to +6y).

- Hence, adding to the very limited number of observations, likely with a low coverage of the extent of the study area, the likely high temporal mismatches with climate and habitat data highlight bid concerns about the reliability of the model outputs.

Specific comments (not an exhausitive list, just the main ones):

l.57-60. This is not anymore true, only three species in the Picoides genus; phylogenetic analyses of nuclear and mitochondrial DNA have shown that the genus Picoides is polyphyletic, and has been restructured (see for instance Shakya, S. B., Fuchs, J., Pons, J. M., & Sheldon, F. H. (2017). Tapping the woodpecker tree for evolutionary insight. Molecular phylogenetics and evolution, 116, 182-191.)

l.61-64. I don’t see the point here to talk about the splitting of Picoides tridactylus in NA, remove this part.

l.72-75. This enumeration is not necessary.

l.164-166. In the study area description, more than one northern limit is cited for the Balkan peninsula, but it is not clear which has been used for the paper.

Figure 1. To understand better the description of the region, it would be helpful to have more indications on the map, such as the Dinarides, or other important features that are mentioned in the study area section.

l.207. Add a reference for the Marko Rakovic field work.

l.209. Add the references to the published literature.

l.214-217. to be deleted.

l.228. ON which based these 4 points were assumed to have imprecise coordinates?

l.280-283. Why not having selected only the most relevant CLC habitat classes for the ATTW as done for the climatic ones?

Table 1 is hard to read and understand. It would be easier if each variable was on a distinct line (aligned for percent contribution and permutation importance), or to have a column for each variable.

Figure 2, the categories for the x-axis in c) and d) should be identified.

In figure 4 and 5, the year (at least) and possible the scenario could be added to the maps. The moderate scenarios could be grouped in a figure and the more extreme scenario in the second figure.

l.570. The mean temperature mentioned (10°C) is not consistent with the temperature mentioned in the results section and in figure 2 (11°C).

L. 559. Incorrect use of i.e.?

L.659-660. Specify that the expected change is a decrease.

L. 574-576. Two ranges of temperature for the mean temperature of the driest quarter and there are 2 peaks for this variable in the figure 2. What could explain it?

Comments on the Quality of English Language

The quality of the English seems relatively ok, but the length of the manuscript is very too long!

Author Response

Dear reviewer,
